# Synthesis, Crystal Structure, Spectral Characterization and Antifungal Activity of Novel Phenolic Acid Triazole Derivatives

**DOI:** 10.3390/molecules28196970

**Published:** 2023-10-07

**Authors:** Pan-Lei Xiao, Xiu-Ying Song, Xin-Ting Xiong, Da-Yong Peng, Xu-Liang Nie

**Affiliations:** 1College of Chemistry & Materials, Jiangxi Agricultural University, Nanchang 330045, China; 2School of Information and Engineering, Jiangxi Agricultural University, Nanchang 330045, China; 3Key Laboratory of Chemical Utilization of Plant Resources of Nanchang, Nanchang 330045, China

**Keywords:** phenolic acid triazole derivatives, synthesis, crystal structure, spectral characterization, antifungal activity

## Abstract

At present, phenolic acid derivatives and triazole derivatives have a good antifungal effect, which has attracted widespread attention. A series of novel phenolic acid triazole derivatives were synthesized, and their structures were characterized by IR, MS, NMR, and X-ray crystal diffraction. Compound methyl 4-(2-bromoethoxy)benzoate, methyl 4-(2-(1*H*-1,2,4-triazol-1-yl) ethoxy)benzoate, 4-(2-(1*H*-1,2,4-triazol-1-yl)ethoxy)benzoic acid and 4-(2-(1*H*-1,2,4-triazol-1-yl) ethoxy)-3-methoxybenzoic acid crystallize in the monoclinic system with space group *P2*_1_*/n*, the monoclinic system with space group *P2*_1_, the monoclinic system with space group *P2*_1_ and the orthorhombic system with space group *Pca2*_1_, respectively. At a concentration of 100 μg/mL and 200 μg/mL, the antifungal activity against seven plant pathogen fungi was determined. Compound methyl 4-(2-bromoethoxy)benzoate has the best inhibitory effect on *Rhizoctonia solani AG1*, and the inhibitory rate reached 88.6% at 200 μg/mL. The inhibitory rates of compound methyl 4-(2-(1*H*-1,2,4-triazol-1-yl) ethoxy)benzoate against *Fusarium moniliforme* and *Sphaeropsis sapinea* at a concentration of 200 μg/mL were 76.1% and 75.4%, respectively, which were better than that of carbendazim.

## 1. Introduction

Plant diseases are one of the most important natural disasters in agricultural production, and plant diseases caused by plant pathogenic fungi have always been the main cause of crop yield reduction [1]. The use of fungicides is the main control measure of agricultural crop diseases and has made great contributions to improving crop yields and quality benefit [2]. However, long-term use of existing traditional fungicides has caused environmental pollution, pesticide residue toxicity, and drug resistance to plant fungal diseases [3]. Therefore, it is of great significance to develop new and promising plant-protection antibacterial agents. A large number of derivatives can be obtained by structural modification of natural products with certain antibacterial activities. It is possible to screen antibacterial functional factors with high activity and little or no side effect, and it is a promising way to develop pesticides and fungicides.

Phenolic acids are substances containing polyphenol hydroxyl groups, including chlorogenic acid, ferulic acid, caffeic acid and rosmarinic acid, which are widely found in natural plants [4,5]. It has pharmacological effects such as anti-oxidative stress [6], improvement of microcirculation [7], improvement of cell energy metabolism [8], etc., and can fully protect the cranial nerves and significantly improve the quality of life of patients with cerebral infarction. Widely used in ischemic cerebrovascular diseases, it has anti-free-radicals and anti-tumor effects [9,10]. Phenolic acid also has a good antibacterial effect [11,12]. However, the further development and utilization of phenolic acids are limited due to their unstable structure, strong hydrophilicity, poor lipid solubility and low bioavailability.

A triazole is a five-membered heterocyclic compound containing three nitrogen atoms. Triazole compounds, as an important part of nitrogen-containing heterocycles, are widely used because of their unique structural characteristics. Triazole compounds have unique biological activity, low toxicity and strong systemic properties. They are often used as structural units of drugs and pesticides, and they play an important role in their synthesis [13,14]. Triazole derivatives have anti-corrosion, antibacterial [15], weeding [16], antivirus [17], anti-influenza [18], anti-cancer [19,20] and anti-oxidation characteristics [21]. Among them, triazole derivatives have a long history of application in sterilization and are an important class of fungicides. Triazole fungicides can be regarded as five-membered heterocyclic derivatives and the structures and positions of the connected atomic groups are different, which leads many triazole fungicides to have different functions (Figure 1) [22,23,24]. Most triazole pesticides are derivatives of a triazole ring substituted by alkyl or aralkyl.

It is known that the combination of two active groups can enhance its activity. For example, Li and Wu, who combined triazoles with pyrimidine, both obtained compounds with good antibacterial activity [25]. Slivka et al. synthesized several compounds with triazole, thiazole and phenol substituents and their copper complexes, and the synthesized compounds had good fungicidal activity [26]. We found that most triazole pesticides are derivatives in which the nitrogen atom of the triazole ring is substituted by alkyl or aralkyl. Because alkylating reagents can easily react with the nitrogen atoms of heterocyclic rings and phenolic hydroxyl groups of phenolic acid, it is very convenient and reasonable to connect triazole rings with phenolic fragments with alkyl chains. We have been committed to the synthesis of and antifungal activity research into natural product derivatives and their complexes, and we have reported a series of crystal structures and biological activity [27,28]. It is found that triazole pesticides have a good inhibitory effect on common crop pathogenic fungi. Therefore, we used 1,2-dibromoethane to introduce 1,2,4-triazole into the phenolic hydroxyl of hydroxybenzoate and used vanillic acid and syringic acid to synthesize a series of phenolic acid triazole derivatives, and we studied their antifungal activities. Herein, we report the synthesis, crystal structure, spectral characterization, and antifungal activity of novel phenolic acid triazole derivatives.

## 2. Results

### 2.1. Chemistry

The synthetic route of compounds **3a–3c** is summarized in Figure 1. The structures of compounds **1a–3c** are shown in Figure 2. All the target phenolic acid triazole derivatives were synthesized by a two-step reaction with methyl *p*-hydroxybenzoate, methyl 4-hydroxy-3-methoxybenzoate and methyl 4-hydroxy-3,5-dimethoxybenzoate as raw materials.

### 2.2. FT-IR Spectral Characterization

All compounds were characterized by an infrared spectrum in the range of 500–4000 cm^−1^. Compounds **1a–1c** are structurally similar, differing only by methoxy in the neighboring position, which is difficult to distinguish in the IR spectrum (Figure 3). **2a–2c** and **3a–3c** are similar to **1a–1c**, so we divided the synthesized compounds into three groups for infrared resolution. **2a–2c** and **3a–3c** show a peak at 3200 cm^−1^, which is consistent with the stretching vibration of N-H on the triazole ring. **3a–3c** shows a broad and scattered peak at 2500–3300 cm^−1,^ consistent with the O-H stretching vibration of the carboxyl group, and a peak near 1700 cm^−1^, consistent with the C=O absorption of the carboxyl group. **1a–1c** and **2a–2c** show a large and deep peak around 1715 cm^−1^, which is consistent with the C=O absorption of the ester group.

### 2.3. NMR Spectral Characterization

In the ^1^H NMR of these compounds, the chemical shifts of the benzene ring and triazole ring are 6.87–8.01 and 7.98–8.60, respectively. The two triplet peaks with the same coupling constants near 4.0 belong to the (-CH_2_CH_2_-) structure. The chemical shift of methyl hydrogen (-CH_3_), connected directly to the O atom, is about 3.85. A unimodal peak above 12.0 is the (-COOH) carboxyl hydrogen.

In the ^13^C NMR of these compounds, the chemical shifts of the carbonyl carbon are in the range of 165.71–166.92. The chemical shift of the triazole ring is in the range of 140.01–152.14. The chemical shifts of the methylene carbon (-CH_2_-) linked to the O atom is in the range of 65.64–72.56, and that of the methyl carbon (-CH_3_) linked to the O atom is in the range of 51.91–56.24. Comparing the (-OCH_3_) substitution of compound **2b** with **2c** in the fifth position, the chemical shifts of **2b** and **2c** in the fourth position are 151 and 140, respectively, the chemical shifts in the fifth position are 112 and 152 and the chemical shifts in the sixth position of the carbon are 122 and 106. The changes in the chemical shifts of the benzene ring carbons of compounds **2b** and **2c** are basically consistent with the above situation.

### 2.4. Crystal Structures of Compounds ***1a–3a*** and ***3b***

#### 2.4.1. Crystal Structure of Compound **1a**


Compound **1a** was crystallized in the monoclinic system with space group *P2*_1_*/n*. In the molecule of **1a** (as shown in Figure 4), the bond lengths and angles were very similar to those of methyl *p*-hydroxybenzoate derivatives [29,30]. The part of methyl *p*-hydroxybenzoate was approximately planar. The dihedral angles of the C1-C6 phenyl plane and the carboxylate group O1-C7-O2 plane were 4.031(187)°. The torsion angles of C4-C7-O2-C8, C1-O3-C9-C10 and O3-C9-C10-Br1 were 178.233(246)°, 144.470(224)° and 69.885(255)°, respectively.

#### 2.4.2. Crystal Structure of Compound **2a**

Compound **2a** was crystallized in the monoclinic system with space group *P2*_1_. In the molecule of **2a** (Figure 5), the methyl *p*-hydroxybenzoate part was approximately planar. The dihedral angles of the C1-C6 phenyl plane, the triazole plane and the carboxylate group O1-C7-O2 plane were 1.190(152)°, 70.285(75)° and 70.283(143)°, respectively. The torsion angles of C6-C7-O2-C8, C3-O3-C9-C10 and O3-C9-C10-N1 were −179.767(185)°, 173.314(157)°, and 69.912(197)°, respectively.

#### 2.4.3. Crystal Structure of Compound **3a**

Compound **3a** was crystallized in the monoclinic system with space group *P2*_1_. In the molecule of **3a** (Figure 6), the methyl *p*-hydroxybenzoate part was approximately planar. The dihedral angles of the C1-C6 phenyl plane, the triazole plane and the carboxylate group O1-C7-O2 plane were 4.767(187)°, 70.984(124)° and 70.009(221)°, respectively. The torsion angles of C3-O3-C9-C10 and O3-C9-C10-N1 were 173.098(219)° and 66.035(289)°, respectively. Intermolecular O–H…N hydrogen bonds between the oxygen atom (O2) of the carboxylate group and the nitrogen atom (N3) of the triazole ring linked the molecules in a one-dimensional chain structure (Figure 7).

#### 2.4.4. Crystal Structure of Compound **3b**

Compound **3b** was crystallized in the orthorhombic system with space group *Pca21*. In the molecule of **3b** (Figure 8), the methyl *p*-hydroxybenzoate part was approximately planar. The dihedral angles of the C2-C7 phenyl plane, the triazole plane and the carboxylate group O1-C1-O2 plane were 3.631(341), 49.059(106) and 50.307(210)°, respectively. The torsion angles of C5-O3-C9-C10, O3-C9-C10-N1 and C5-C6-O4-C8 were −174.442(232), 66.341(300) and −177.680(259)°, respectively. Intermolecular O–H…N hydrogen bonds between the oxygen atom (O2) of the carboxylate group and the nitrogen atom (N3) of the triazole ring linked the molecules into a one-dimensional chain structure (Figure 9).

### 2.5. Antifungal Activity Analysis

The diameter of each inhibition zone (IZ) was measured to calculate the inhibition rates of the compounds. The results listed in Table 1 show that compounds **1a–2c** were effective against these seven plant pathogenic fungi: *Rhizoctonia solani. AG1*, *Fusarium moniliforme*, *Colletotrichum fructicola*, *Phytophthora parasitica* var. *nicotianae*, *Fusarium oxysporum* f. sp. *niveum*, *Fusarium verticillioide* and *Sphaeropsis sapinea*. Among them, compound **1a** has the strongest inhibitory activity against *Rhizoctonia solani. AG1*, *Phytophthora parasitica* var., *nicotine*, *Fusarium oxysporum* f. sp. *niveum* and *Fusarium verticillioide*. At a concentration of 200 μg/mL, the inhibitory rate against *Rhizoctonia solani. AG1* was the highest, reaching 88.6%. For structurally similar compounds, compounds **1a–1c** showed similar inhibitory activity against *Colletotrichum fructicola*, with the remaining six fungi all following the **1a** ˃ **1b** ˃ **1c** pattern. This may be due to the methoxylation (-OCH_3_) of the neighboring position, so we make a guess: for **1a**, the electron-donating group in the neighboring position diminishes its inhibitory activity. Although compound **1a** was less effective than the positive control carbendazim, we believe it has the potential to synthesize compounds with good inhibition. Specifically, the inhibitory rates of compound **1a** against *Rhizoctonia solani. AG1* and *Phytophthora parasitica* var. *nicotianae* were 88.6% and 80.7%, respectively (on the left of Figure 10). Among them, compound **2c** has the highest inhibitory effect on *Rhizoctonia solani. AG1*, and the inhibitory rate was 56.8% at a concentration of 200 μg/mL. Compound **2b** has the highest inhibitory effect on *Colletotrichum fructicola* and the inhibitory rate was 45.1% at a concentration of 200 μg/mL. Compound **2a** has a better inhibitory effect on *Fusarium moniliforme* and *Sphaeropsis sapinea* than other compounds. The inhibitory rates of compound **2a** against *Fusarium moniliforme* and *Sphaeropsis sapinea* at 200 μg/mL were as high as 76.1% and 75.4%, respectively (on the right of Figure 10), which was better than those of carbendazim. Compounds **3a–3c** have poor antifungal activity against seven plant pathogenic fungi, which may be due to their high hydrophilicity and low lipid solubility.

## 3. Discussion

From the perspective of inhibitory effects, we think that compounds **1a** and **2a** have good antifungal potential. Compound **1a** showed good inhibitory activity against all seven tested fungi. It contained bromine and halogen, which are not uncommon in pesticide fungicides, such as iprodione, dimethachlon and bromonirol. In the future, we will consider adding some halogen to the compounds with better antifungal activity. In the analysis of our results against fungi, we compared **1a**, **1b** and **1c** and found that the electron-donating group in the ortho position may reduce the activity of **1a**. We will try to introduce an electron-withdrawing group in the ortho position to test whether this is consistent with our idea. Compound **2a** has a very good inhibitory effect on *Fusarium moniliforme* and *Sphaeropsis sapinea*, even better than that of carbendazim. Compared with triazole fungicides such as triadimefon and phenoxymethoxazole, compound **2a** has a simple structure and is easy to modify. We will also try to modify **2a** in the future, possibly by introducing halogen or modifying the methyl ester into an amide structure.

## 4. Experimental Procedure

### 4.1. Materials and Methods

FT-IR spectrum at 500–4000 cm^−1^ was confirmed by potassium bromide compression method. ^1^H NMR and ^13^C were recorded on the NMR spectrum of Bruker Advance NEO 500 M instrument, TMS was used as the internal standard and CDCl_3_ or DMSO-d6 was used as the internal solvent. X-ray analysis of the crystal samples was measured by graphite monochromatic Mo Kα radiation (*λ* = 0.71073 Å) under the diffraction instrument of Bruker Apexii detector. Melting point is determined by X-4 microscopic melting point instrument (Shanghai Jingmi Co., Ltd., Shanghai, China) without correction. 

Unless otherwise specified, all chemical reagents are commercially available and have been treated with standard methods before use. The progress of the chemical reactions and the purity of all products were monitored with thin-layer chromatography (TLC) on silica gel plates (silicone 60 GF254, ready-to-use). All strains of plant pathogenic fungi were provided by the Plant Pathology Laboratory of the College of Agricultural, Jiangxi Agricultural University. The cultivation bases come from Beijing Boxing Biotechnology Co., Ltd., Beijing, China.

### 4.2. Synthesis of ***3a–3c***

#### 4.2.1. Synthesis of Compounds **1a–1c**

Compounds **1a–1c** were synthesized by the revised method of reference [28].

#### 4.2.2. Synthesis of Compounds **2a–2c**

In the synthesis example, compounds **2a, 2b** and **2c** were synthesized in the same way, except that **1a** was replaced by **1b** and **1c**. Methyl 4-(2-bromoethoxy)benzoate (**1a**) (0.05 mol, 12.96 g) and sodium triazole (0.05 mol, 4.55 g) were dissolved in acetonitrile (30 mL). The reaction mixture was stirred well for about 5 h at 85 °C. Reaction was monitored by TLC. After the reaction was completed, the obtained solution was filtered and washed with water (10 mL) three times. The filtrate was extracted with Ethyl acetate and dried over anhydrous Na_2_SO_4_. The product was purified to obtain compounds **2a–2c** by column chromatography on silica gel using ethyl acetate/petroleum ether (V_Ethyl acetate_/V_petroleum ether_ = 3:1) as eluent.

#### 4.2.3. Syntheses of Compounds **3a–3c**

For example, **3a** was synthesized in the same way as **3b** and **3c**, except that **2b** and **2c** replaced **2a**. The mixture of methyl 4-(2-(1*H*-1,2,4-triazol-1-yl) ethoxy) benzoate (**2a**) (0.01 mol, 2.47 g), NaOH solution (10%, 40 mL) and ethanol (2 mL) was reacted under stirring at room temperature. Reaction was monitored by TLC. After the reaction was completed, the pH of the obtained solution was adjusted to 5–6 with diluted hydrochloric acid. A large amount of white solids was precipitated, filtered and dried, and the products **3a–3c** were obtained. The crystals were obtained after one week of slow volatilization at room temperature.
*Methyl 4-(2-bromoethoxy) benzoate* (**1a**): white solid; the yield is 35.8%, m. p: 63.4~65.0 °C. ^1^H NMR (500 MHz, CDCl_3_) δ 8.00 (d, *J* = 8.8 Hz, 2H), 6.93 (d, *J* = 8.8 Hz, 2H), 4.34 (t, *J* = 6.2 Hz, 2H), 3.89 (s, 3H), 3.66 (t, *J* = 6.2 Hz, 2H). ^13^C NMR (126 MHz, CDCl_3_) δ 166.91, 161.99, 131.90, 123.55, 114.43, 68.04, 52.13, 28.82. HRMS: C_10_H_11_BrO_3_ for [M + H]^+^, calculated 258.9925, found 258.9960.*Methyl 4-(2-bromoethoxy)-3-methoxybenzoate* (**1b**): white solid; the yield is 40.2%, m. p: 69.6~72.1 °C. ^1^H NMR (500 MHz, CDCl_3_) δ 7.65 (d, *J* = 10.3 Hz, 1H), 7.57 (s, 1H), 6.90 (d, *J* = 8.4 Hz, 1H), 4.38 (t, *J* = 6.7 Hz, 2H), 3.92 (s, 3H), 3.90 (s, 3H), 3.68 (t, *J* = 6.7 Hz, 2H). ^13^C NMR (126 MHz, CDCl_3_) δ 166.68, 151.39, 149.13, 123.75, 123.26, 112.88, 112.63, 68.75, 56.14, 52.04, 28.29. HRMS: C_11_H_13_BrO_4_ for [M + H]^+^, calculated 289.0031, found 289.0072.*Methyl 4-(2-bromoethoxy)-3,5-dimethoxybenzoate* (**1c**): white solid; the yield is 36.7%, m. p: 102.1~103.5 °C. ^1^H NMR (500 MHz, CDCl_3_) δ 7.29 (s, 2H), 4.34–4.29 (m, 2H), 3.93 (s, 3H), 3.92 (s, 6H), 3.63–3.58 (m, 2H). ^13^C NMR (126 MHz, CDCl_3_) δ 166.58, 152.91, 140.35, 125.69, 106.69, 72.56, 56.24, 52.25, 29.47. HRMS: C_12_H_15_BrO_5_ for [M + H]^+^, calculated 319.0136, found 319.0170.*Methyl 4-(2-(1H-1,2,4-triazol-1-yl) ethoxy) benzoate* (**2a**): white solid; the yield is 62.5%, m. p: 113.5~116.1 °C. ^1^H NMR (500 MHz, CDCl_3_) δ 8.22 (d, *J* = 9.0 Hz, 2H), 7.98 (s, 1H), 7.97 (d, *J* = 3.2 Hz, 1H), 6.87 (d, *J* = 8.8 Hz, 2H), 4.60 (t, *J* = 5.0 Hz, 2H), 4.39 (t, *J* = 5.0 Hz, 2H), 3.88 (s, 3H). ^13^C NMR (126 MHz, CDCl_3_) δ 166.82, 161.43, 152.14, 143.96, 131.69, 123.58, 114.06, 65.64, 51.95, 49.07. HRMS: C_12_H_13_N_3_O_3_ for [M + H]^+^, calculated 248.0990, found 248.1028.*Methyl 4-(2-(1H-1,2,4-triazol-1-yl)ethoxy)-3-methoxybenzoate* (**2b**): white solid; the yield is 65.6%, m. p:118.3~120.4 °C. ^1^H NMR (500 MHz, DMSO) δ 8.55 (s, 1H), 7.99 (s, 1H), 7.55 (d, *J* = 8.4 Hz, 1H), 7.45 (s, 1H), 7.08 (d, *J* = 8.5 Hz, 1H), 4.62 (t, *J* = 5.1 Hz, 2H), 4.43 (t, *J* = 5.1 Hz, 2H), 3.81 (s, 3H), 3.78 (s, 3H). ^13^C NMR (126 MHz, DMSO) δ 165.83, 151.50, 151.42, 148.57, 144.63, 122.99, 122.50, 112.70, 112.23, 66.54, 55.71, 51.91, 48.23. HRMS: C_13_H_15_N_3_O_4_ for [M + H]^+^, calculated 278.1096, found 278.1125.*Methyl 4-(2-(1H-1,2,4-triazol-1-yl)ethoxy)-3,5-dimethoxybenzoate* (**2c**): white solid; the yield is 59.8%, m. p: 88.1~89.6 °C. ^1^H NMR (500 MHz, DMSO) δ 8.60 (s, 1H), 8.04 (s, 1H), 7.27 (s, 2H), 4.54 (t, *J* = 5.0 Hz, 2H), 4.37 (t, *J* = 5.0 Hz, 2H), 3.91 (s, 3H), 3.84 (s, 6H). ^13^C NMR (126 MHz, DMSO) δ 165.71, 152.53, 151.15, 144.62, 140.01, 124.98, 106.22, 70.52, 55.95, 52.18, 49.24. HRMS: C_14_H_17_N_3_O_5_ for [M + 1], calculated 308.1202, found 308.1215.*4-(2-(1H-1,2,4-triazol-1-yl) ethoxy) benzoic acid* (**3a**): white solid; the yield is 84.8%, m. p: 194.3~197.5 °C. ^1^H NMR (500 MHz, DMSO) δ 12.64 (s, 1H), 8.58 (s, 1H), 7.99 (s, 1H), 7.87 (d, *J* = 8.7 Hz, 2H), 7.00 (d, *J* = 8.8 Hz, 2H), 4.61 (t, *J* = 5.0 Hz, 2H), 4.43 (t, *J* = 5.0 Hz, 2H). ^13^C NMR (126 MHz, DMSO) δ 166.84, 161.41, 151.49, 144.68, 131.31, 123.41, 114.31, 65.87, 48.18. HRMS: C_11_H_11_N_3_O_3_ for [M + H]^+^, calculated 234.0834, found 234.0859.*4-(2-(1H-1,2,4-triazol-1-yl) ethoxy)-3-methoxybenzoic acid* (**3b**): white solid; the yield is 82.1%, m. p: 178.3~180.2 °C. ^1^H NMR (500 MHz, DMSO) δ 12.70 (s, 1H), 8.54 (s, 1H), 7.98 (s, 1H), 7.52 (d, *J* = 10.2 Hz, 1H), 7.44 (s, 1H), 7.05 (d, *J* = 16.2 Hz, 1H), 4.62 (t, *J* = 5.1 Hz, 2H), 4.43 (t, *J* = 5.1 Hz, 2H), 3.77 (s, 3H). ^13^C NMR (126 MHz, DMSO) δ 166.92, 151.39, 151.15, 148.49, 144.60, 123.74, 123.01, 112.67, 112.53, 66.54, 55.64, 48.24. HRMS: C_12_H_13_N_3_O_4_ for [M + H]^+^, calculated 264.0940, found 264.0946.*4-(2-(1H-1,2,4-triazol-1-yl) ethoxy)-3,5-dimethoxybenzoic acid* (**3c**): white solid; the yield is 80.2%, m. p: 168.5~171.7 °C. ^1^H NMR (500 MHz, DMSO) δ 8.54 (s, 1H), 7.96 (s, 1H), 7.19 (s, 2H), 4.46 (t, *J* = 5.0 Hz, 2H), 4.28 (t, *J* = 5.0 Hz, 2H), 3.74 (s, 6H). ^13^C NMR (126 MHz, DMSO) δ 167.05, 152.42, 151.19, 144.70, 139.49, 127.01, 106.37, 70.55, 55.92, 49.34. HRMS: C_13_H_15_N_3_O_5_ for [M + H]^+^, calculated 294.1045, found 294.1055. 


### 4.3. Crystal Structure Determination

The crystal structures of **1a–3a** and **3b** were determined by X-ray single-crystal diffraction (see Appendix A). Reflectance data were collected at room temperature using Bruker APEX II area detector [31], which was equipped with a graphite-monochromatic MoKα radiation (*λ* = 0.71073 Å) at 296(2) K, and the scanning mode was *ω*-2*θ*. All data were corrected using SAD ABS by empirical adsorption. The structures were solved by direct methods and refined by full-matrix least squares on *F*^2^ using SHELXTL 97 software [32,33]. All non-hydrogen atoms were located by the direct method and subsequent differential Fourier synthesis method.

Crystallographic data for the structures described in this paper were stored in the Cambridge Crystallographic Data Centre (CCDC) with identification numbers 2214417 (**1a**), 2214418 (**2a**), 2214429 (**3a**) and 2214420 (**3b**).

### 4.4. Antifungal Activity Test

The antifungal activity of the target compounds and their intermediates against 7 plant pathogenic fungi were determined by the disk diffusion method [34,35] at 100 μg/mL and 200 μg/mL. We used Carbendazim as a bactericide comparison. 

The preparation of the test sample: 20 mg of the test sample was dissolved in 1 mL of acetone, then the solution was transferred to a 10 mL volumetric flask, with 0.1% tween-80 aqueous solution used for constant volume. The prepared solution was poured into 90 mL (50~55 °C) potato glucose agar (PDA) medium, mixed evenly, and then quickly poured into 6 Petri dishes (90 mm in diameter) to detect 2 kinds of fungus. At the same time, a blank control group (0.1% tween-80 aqueous solution) without drugs was set up. 

Inoculation of pathogenic fungus: the fungus was cut from the edge of the colony with a sterile drill (5 mm) and was inoculated in the center of the dish containing the medicine with an inoculation stick. Then, the Petri dishes were cultured in an illumination incubator (28 °C) for 5–7 days. When the hyphae of the fungi in blank control group grew to 7–8 cm, all experimental data of the fungi were recorded. The diameter of the mycelium was measured using the cross method. Each antifungal test was repeated separately three times. The inhibition ratio *I* (%) was calculated by the following formula, in which *C* represents the diameter of control group (untreated with compound) and *T* represents the diameter of treated group [36]:(1)$$I(\%)=\frac{\left(C-T\right)}{\left(C-0.5\right)}\times 100.$$

## 5. Conclusions

Phenolic acid triazole derivatives were synthesized by introducing a triazole ring into the hydroxyl group of phenolic acid through the alkyl chain. The structures of all the target compounds were characterized by IR, MS and NMR. Single crystals were cultured with synthetic compounds, and compound methyl 4-(2-bromoethoxy) benzoate, methyl 4-(2-(1*H*-1,2,4-triazol-1-yl) ethoxy) benzoate, 4-(2-(1*H*-1,2,4-triazol-1-yl) ethoxy)benzoic acid and 4-(2-(1*H*-1,2,4-triazol-1-yl)ethoxy)-3-methoxybenzoic acid were cultured with X-ray crystal. The potential antifungal activity against seven plant pathogens at 100 μg/mL and 200 μg/mL were tested. Research has shown that the compounds have antifungal activity against plant pathogens. Compound methyl 4-(2-bromoethoxy) benzoate has a good inhibitory effect on the tested fungi. Among them, compound methyl 4-(2-bromoethoxy) benzoate has the highest inhibitory rate against *Rhizoctonia solani AG1*, which reaches 88.6% at 200 μg/mL. The inhibitory rates of compound methyl 4-(2-(1*H*-1,2,4-triazol-1-yl) ethoxy)benzoate against *Fusarium moniliforme* and *Sphaeropsis sapinea* at 200 μg/mL were 76.1% and 75.4%, respectively, which were superior to those of carbendazim.

## Data Availability

CCDC numbers 2214417 (**1a**), 2214418 (**2a**), 2214429 (**3a**), and 2214420 (**3b**) contain the supplementary crystallographic data for the structures reported in this paper. Copies of this information may be obtained free of charge from the Director, CCDC, 12 Union Road, Cambridge CB2 1EZ, UK (Fax: (+44) 1223 336-033; e-mail: deposit@ccdc.cam.ac.uk; or http://www.ccdc.cam.ac.uk, accessed on 1 August 2023) or also available from the author Xu-Liang Nie. accessed on 1 January 2025.

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
