# Peer review of "Synthesis, Crystal Structure, Spectral Characterization and Antifungal Activity of Novel Phenolic Acid Triazole Derivatives"

_molecules, 2023, doi:10.3390/molecules28196970_

Round 1
Reviewer 1 Report (Previous Reviewer 2)
One of the primary concerns raised in the initial review was the lack of clarity in the article's structure. The organization of the content made it difficult for readers to follow the logical flow of the argument. Unfortunately, this concern remains unaddressed in the final version of the article.
The manuscript continues to have the same deficiencies and does not have the necessary quality to be published in this journal, so it should not be accepted for publication.
English usage in this manuscript must be substantially improved. There are many grammatical errors and vague descriptions.
Author Response
According to your revision, I adjusted the structure of the paper, and carefully revised the organization, logic and grammar of the content. After the revision, the quality of the manuscript was greatly improved.
Reviewer 2 Report (New Reviewer)
The manuscript "Synthesis, Crystal Structure, Spectroscopic Characterization and Antifungal Activities of Novel Phenolic Acids Triazole Derivatives", reports important effects antifungal of Novel Phenolic Acids Triazole Derivatives. However, the authors should clarify the following points:
· The authors should rewrite the abstract since in it they suggest the anti-cancer and antioxidant properties of the compounds, when in fact only the antifungal activity was determined. This causes confusion in readers.
· The authors must add in the material and methods section (antifungal tests section) the fungi used. Likewise, the authors must indicate what was the criteria for selecting these pathogenic fungi?
· Why weren't in vitro cytotoxicity tests performed to determine the safety of the compounds?
· The authors must review the entire manuscript since there are typographical errors and spaces.
The authors must review the entire manuscript, since the new version presents fragments written in British English and fragments in American English. They should make the entire manuscript uniform.
Author Response
We are grateful for your many constructive comments and suggestions.
- We have rewritten the summary section.
- we have added (antifungal tests section) the fungi used in the material and methods section.
- We choose these pathogenic fungi because they are the most common pathogenic fungi in crop production.
- Thank you for your advice. Next, I am going to do the anti-tumor activity and in vitro toxicity test of these compounds.
- We have carefully revised the writing and grammar of the entire manuscript.
Reviewer 3 Report (New Reviewer)
Comments to paper:
Synthesis, Crystal Structures, Spectroscopic Characterization and Antifungal Activity of Novel Phenolic Acids Triazole Derivatives
by Pan-Lei Xiao, Xiu-Ying Song, Xin-Ting Xiong, Da-Yong Peng, Xu-Liang Nie
The derivatives of azoles have found wide applications in different fields of chemistry, technology, medicine, and agriculture. Various therapeutic products, herbicides, pesticides, and plant growth regulators have been created from them. The interest in azole derivatives, in particular, 1,2,4-triazoles is constantly growing. The presence of unusual properties of the triazole ring, such as high chemical stability, a large dipole moment, tautomeric rearrangements, heteroaromatic nature of the cycle and the ability to form hydrogen bonds are effectively used in fine organic synthesis for the construction of biologically active molecules and new materials. Triazoles have unique biological activity, low toxicity and high systemicity. Some triazoles have a long history of use in sterilization and are an important class of fungicides.
The authors synthesized novel phenolic acid triazoles, carried out their structural study by IR, MS, NMR and X-ray crystal diffraction methods and analysis for antifungal activity.
In my opinion, the work is very interesting, original, and relevant, performed on up-to-date theme and has significant practical application. I think that the article may be of interest both to specialists in this field and to a wide audience of the scientific community.
The authors skillfully used NMR spectroscopy. The entire experiment was carried out at the expert level and the manuscript was written professionally, which prompts me to recommend this article for publication in the journal of Molecules.
However, I would like to point out the following. As wishes for the future, I would like to see the assignment of NMR signals, in particular, 13C signals. You have such a wonderful Bruker spectrometer at 500 MHz and it’s not very professional to give signals without assignment.
Comments to paper:
Synthesis, Crystal Structures, Spectroscopic Characterization and Antifungal Activity of Novel Phenolic Acids Triazole Derivatives
by Pan-Lei Xiao, Xiu-Ying Song, Xin-Ting Xiong, Da-Yong Peng, Xu-Liang Nie
The derivatives of azoles have found wide applications in different fields of chemistry, technology, medicine, and agriculture. Various therapeutic products, herbicides, pesticides, and plant growth regulators have been created from them. The interest in azole derivatives, in particular, 1,2,4-triazoles is constantly growing. The presence of unusual properties of the triazole ring, such as high chemical stability, a large dipole moment, tautomeric rearrangements, heteroaromatic nature of the cycle and the ability to form hydrogen bonds are effectively used in fine organic synthesis for the construction of biologically active molecules and new materials. Triazoles have unique biological activity, low toxicity and high systemicity. Some triazoles have a long history of use in sterilization and are an important class of fungicides.
The authors synthesized novel phenolic acid triazoles, carried out their structural study by IR, MS, NMR and X-ray crystal diffraction methods and analysis for antifungal activity.
In my opinion, the work is very interesting, original, and relevant, performed on up-to-date theme and has significant practical application. I think that the article may be of interest both to specialists in this field and to a wide audience of the scientific community.
The authors skillfully used NMR spectroscopy. The entire experiment was carried out at the expert level and the manuscript was written professionally, which prompts me to recommend this article for publication in the journal of Molecules.
However, I would like to point out the following. As wishes for the future, I would like to see the assignment of NMR signals, in particular, 13C signals. You have such a wonderful Bruker spectrometer at 500 MHz and it’s not very professional to give signals without assignment.
Author Response
We are grateful for your many constructive comments and suggestions.
We rewrote 2.2 NMR spectral characterization.
We reassigned the give signals.
Reviewer 4 Report (New Reviewer)
General information about the biological activity of triazoles (line 46) should be accompanied by an appropriate link to a review work (for example, European Journal of Medicinal Chemistry. 2015. Vol. 97. pp. 830-870 or European Journal of Medicinal Chemistry. 2019. Vol. 165. 332-346 or other).
When justifying the choice of the research object (line 61-65), in addition to mentioning the synergistic biological effect of the pyrimidine fragment and triazole, it is also worth mentioning the fungicidal effect (just on plant pathogens) of compounds with a combination of triazole, thiazole and a phenolic substituent in the triazole heterocycle ( Letters in Drug Design and Discovery, 2022, 19(9), pp. 791–799). And then to describe the expediency of studying objects in which the triazole ring and the phenolic fragment are separated by an alkyl chain. It should also be justified to choose an alkylating agent (arylsibromethane), which works well in the alkylation of the nitrogen atom of other heterocycles (for example, Pharm Chem J 29, 750–752 (1995). https://doi.org/10.1007/BF02331852 or Pharm Chem J 40, 485–488 (2006). https://doi.org/10.1007/s11094-006-0160-1 ).
A lot of information on synthesis methods and analysis data (when proving structures) should be placed in supporting material and removed from the main text:
a) compounds 1 are known and are essentially commercial reagents, so their detailed procedure of synthesis should not be given in the manuscript. And the XRD record for the known compound (1a) looks completely strange!... it should be removed.
b) most of the digital data of the XRD (for example, tables 2, 3) should also be given in the supporting material, and only the corresponding conclusions on the structure should be left in the main text.
The recording frequencies of NMR spectra on the Bruker Avance NEO 500 M device should be listed separately for 13С and for 1Н (line 197), and not to duplicate and repeat this information for each compound.
Numbering of compounds in the abstract and conclusions should be avoided.
It seems strange to study the fungicidal activity exclusively only for plant pathogens. This should be further substantiated. Or add fungicidal studies on at least one of the classic pathogens, such as Candida.
In the supporting material, data from cif files for XRD should be provided to assess the quality and reliability of the given data (after all, XRD is provided for structures that do not contain reference heavy atoms!)
Average quality of English
Author Response
We thank you for your many constructive comments and suggestions, which are very helpful in improving the quality of the manuscript.
According to your suggestion, I revised the Introduction and replaced two references, revised the Abstract and Conclusions.
In this manuscript, we have studied the common pathogenic fungi. According to your suggestion, we will carry out bacteria test next time.
According to your suggestion, I deleted Table 3. We provided the data from cif files for XRD.
.
Reviewer 5 Report (New Reviewer)
I would only recommend that the authors review the scientific names, these should be in italics, for example line 15, 149.
Author Response
Thank you very much for your suggestion, and we have made corresponding changes.
Round 2
Reviewer 1 Report (Previous Reviewer 2)
The manuscript still exhibits the same deficiencies, and I have not received any responses to my previous comments. For instance,
1. Figure 1 is of poor quality and difficult to understand.
2. The Scheme 1 should be improved, add missing conditions, and use superscript - instead “Rn” should be “Rn”.
3. The Results section should begin with the synthesis of the title compounds.
Furthermore, could the authors please provide an explanation for the presence of a significant number of additional signals in the NMR spectra? Additionally, it would be helpful if the authors could describe the methods they used to confirm the purity of the compounds they obtained and provide the degree of purity achieved.
There are many typos/inaccuracies in the document, such as "All reagents and solvents are reagent or purified by purifying the use of silicone 60 GF254 200 (China Co., Ltd.) for thin layer chromatography (TLC) and preparing thin layer chromatography (PTLC)."
Author Response
The manuscript still exhibits the same deficiencies, and I have not received any responses to my previous comments. For instance,
- Figure 1 is of poor quality and difficult to understand.
Response: Thank you for your valuable advice. We have revised the Figure 1 and revised the caption of Figure 1.
- The Scheme 1 should be improved, add missing conditions, and use superscript - instead “Rn” should be “Rn”.
Response: Thank you for your careful work. We have revised the Scheme 1, and the reaction conditions are given in the illustration.
- The Results section should begin with the synthesis of the title compounds.
Response: Thank you for your valuable advice. We have revised the Results section
Furthermore, could the authors please provide an explanation for the presence of a significant number of additional signals in the NMR spectra? Additionally, it would be helpful if the authors could describe the methods they used to confirm the purity of the compounds they obtained and provide the degree of purity achieved.
Response: Thank you for your careful work. All the synthetic products were confirmed to be correct by IR, NMR and single crystal diffraction techniques, so the additional signals were not explained. The separation and purification method of products and intermediates is added to the Materials and Methods part.
There are many typos/inaccuracies in the document, such as "All reagents and solvents are reagent or purified by purifying the use of silicone 60 GF254 200 (China Co., Ltd.) for thin layer chromatography (TLC) and preparing thin layer chromatography (PTLC)."
Response: Thank you for your careful work. We have revised many typos/inaccuracies in the document.
Reviewer 4 Report (New Reviewer)
My comments were only partially processed by the authors! The manuscript needs additional editing!
In particular:
1. The authors ignored the proposal to justify the choice of the alkylating reagent, and also left in the manuscript a detailed description of the preparation of the known compounds (1) with X-rays for (1a) - the corresponding corrections in the manuscript are necessary!
2. The authors left detailed digital X-ray data in the main text, which clutters the main content - it must be transferred to supplementary materials; instead, the manuscript should include information about the deposit (CCDC number) of these studies with a direct link to the Cambridge Crystallographic Data centre site (http://www.ccdc.cam.ac.uk/)!. It is also necessary to add the cif-file of the X-ray to the supplementary materials.
3. It is not necessary to duplicate the frequencies of NMR recording on 1H and 13C nuclei for each compound – this information is in the general description.
4. Numbering of compounds in the abstract and conclusions should be avoided..
5. If the authors do not plan to expand the set of fungi in this manuscript, then the rationale for choosing the object of bio-research should be given in the article!... Why exactly were plant’ pathogens studied?
In addition to everything else - a lot of grammatical errors can be found in the revised text! – it is necessary to check the English language in the final version of the manuscript!
А lot of grammatical errors can be found in the revised text! – it is necessary to check the English language in the final version of the manuscript!
Author Response
My comments were only partially processed by the authors! The manuscript needs additional editing!
In particular:
- The authors ignored the proposal to justify the choice of the alkylating reagent, and also left in the manuscript a detailed description of the preparation of the known compounds (1) with X-rays for (1a) - the corresponding corrections in the manuscript are necessary!
Response: Thank you for your advice. We have increase that rationality of the selection of alkylating agent, simplified the description of the preparation of compound (1).
- The authors left detailed digital X-ray data in the main text, which clutters the main content - it must be transferred to supplementary materials; instead, the manuscript should include information about the deposit (CCDC number) of these studies with a direct link to the Cambridge Crystallographic Data centre site (http://www.ccdc.cam.ac.uk/)!. It is also necessary to add the cif-file of the X-ray to the supplementary materials.
Response: Thank you for your advice. We have deleted all tables of single crystal data, and added the information about the deposit (CCDC number).
It is not necessary to duplicate the frequencies of NMR recording on 1H and 13C nuclei for each compound – this information is in the general description.
Response: Thank you for your careful work. According to your suggestion,we have revised here.
- Numbering of compounds in the abstract and conclusions should be avoided.
Response: Thank you for your careful work. According to your suggestion,we have revised the abstract and conclusions.
- If the authors do not plan to expand the set of fungi in this manuscript, then the rationale for choosing the object of bio-research should be given in the article!... Why exactly were plant’ pathogens studied?
Response: Thank you for your valuable advice. We are in agricultural university, and the research object is generally the fungicidal activity of plant pathogens. According to your suggestion, we will study the fungicides of other typical pathogens (such as Candida) in the future. In addition, we also bought some bacterial strains, and we will also carry out anti-bacteria test in the future.
In addition to everything else - a lot of grammatical errors can be found in the revised text! – it is necessary to check the English language in the final version of the manuscript!
Response: Thank you for your careful work. We have revised many grammatical errors in the document.
Round 3
Reviewer 1 Report (Previous Reviewer 2)
The manuscript still exhibits the same deficiencies, and I have not received any responses to my previous comments. For instance,
1) The Results section should begin with the synthesis of the title compounds.
I attached the manuscript with comments and notes, please, carefully read and correct all the points.

English usage in this manuscript must be substantially improved.
Author Response
Third round Reviewer 1
The manuscript still exhibits the same deficiencies, and I have not received any responses to my previous comments. For instance,
- The Results section should begin with the synthesis of the title compounds.
Response: According to your valuable advice, we adjusted the structure of the paper. Now the Results section begin with the synthesis of the title compounds. See “2.1 Chemistry” for details.
I attached the manuscript with comments and notes, please, carefully read and correct all the points.
Response: Thank you for your careful work. We carefully corrected all the spelling mistakes and formatting errors you marked.
We also checked the full text, and we corrected some grammatical errors. After the revision, the quality of the manuscript was greatly improved.
Reviewer 4 Report (New Reviewer)
In general, my comments were taken into account. The current version of the manuscript meets the requirements and does not contain gross blunders, so it can be published.
English is acceptable
Author Response
Third round Reviewer 4
- In general, my comments were taken into account. The current version of the manuscript meets the requirements and does not contain gross blunders, so it can be published.
Response: Thank you very much for your affirmation of our work. We checked the full text carefully, and we corrected some grammatical errors. After the revision, the quality of the manuscript was greatly improved.
Round 4
Reviewer 1 Report (Previous Reviewer 2)
The manuscript may be accepted after making some corrections in accordance with the comments.
Figure 7: should be not bold.
Line 268 "3.91 (s, J = 12.3 Hz, 6H)" and line 273 "3.90 (s, J = 5.4 Hz, 9H)": singlets with J-coupling! Could you please be more attentive to details, it's not professional to make such mistakes!
Line 348: "against" instead of "on"
Author Response
Fourth round Reviewer 1
The manuscript may be accepted after making some corrections in accordance with the comments.
Figure 7: should be not bold.
Response: Thank you for your careful work. We corrected it.
Line 268 "3.91 (s, J = 12.3 Hz, 6H)" and line 273 "3.90 (s, J = 5.4 Hz, 9H)": singlets with J-coupling! Could you please be more attentive to details, it's not professional to make such mistakes!
Response: Thank you for your careful work. We corrected it.
Line 348: "against" instead of "on"
Response: Thank you for your careful work. We corrected it.
This manuscript is a resubmission of an earlier submission. The following is a list of the peer review reports and author responses from that submission.
Round 1
Reviewer 1 Report
The authors of the manuscript "Synthesis, crystal structure, spectroscopic characterization and antifungal activity of phenolic acids triazole derivatives", the main contribution is the antifungal activity of these compounds. This work was submitted in February in this same journal and was rejected due to deficiencies in the writing and incorrect description of the trials, mainly the antifungals that are the central part of this work, among other deficiencies that the manuscript presented.
The authors responded to the questions about the antifungal trials, leaving more doubts about the trials and the results obtained.
The current manuscript mentions the use of pathogenic bacteria: "The cultured pathogenic bacterium were then cut from the edge of the colony under aseptic conditions using a 5 mm diameter sterilized punch"; isn't it antifungal?
The formula used to determine the % inhibition of mycelial growth is incorrect (its use would give negative values or values greater than 100%), so how can you be certain of the data obtained?
In addition, they did not show evidence of the antifungal tests with the good activity of the compounds "Thank you very much for your advice, we did not take any photos at the time of the error and the strain used for the experiment needs to be activated, please give us some time to add this aspect".
The authors limited themselves to making specific corrections but not in-depth, with unacceptable errors, for example. "Use Carbendazim as a bactericide comparison" Carbenzamin is a compound with broad-spectrum antifungal activity, or the change of bacteria in the tests. The document continues to have the same deficiencies and does not have the necessary quality to be published in this journal, so it should not be accepted for publication.
Minor editing of English language required
Reviewer 2 Report
The authors report the synthesis, crystal structure, spectroscopic characterization, and antifungal activity of phenolic acids triazole derivatives.
The manuscript is poorly structured, has incorrect English grammar, many errors and typos, it is often impossible to understand the text. The submitted manuscript cannot be accepted for publication in Molecules. I carefully checked the points of the previous reviewers and authors' responses. In many cases the answers are insufficient and poor.
1. What does “MRI” mean?
2. The Results section should begin with the synthesis of the title compounds.
3. Figure 1 is of poor quality and difficult to understand.
4. It is inappropriate to compare experimental spectra with Chembiodraw simulations for publication.
5. You should add explanation to Table 3, what does “V” mean? You should add values for concentration, the formula for inhibition rate.
6. Why are ppm values used instead of standard μg/mL?
7. The Scheme 1 should be improved, add missing conditions, and use superscript - instead “Rn” should be “Rn”.
8. Specify the eluents for column chromatography.
9. In Conclusion you state: “Among them, the compound 1a has the highest inhibitory rate of the eggplant nucleus, which can reach 88.6% at 200 ppm.” But there is no mention of eggplant nucleus in the text.
Because of the abovementioned serious problems, I suggest the manuscript for Reject.
English usage in this manuscript must be substantially improved. There are many grammatical errors and vague descriptions.